# Comparison between Fourth-Generation FloTrac/Vigileo System and Continuous Thermodilution Technique for Cardiac Output Estimation after Time Adjustment during Off-Pump Coronary Artery Bypass Graft Surgery: A Retrospective Cohort Study

**DOI:** 10.3390/jcm11206093

**Published:** 2022-10-16

**Authors:** Chahyun Oh, Soomin Lee, Pyeonghwa Oh, Woosuk Chung, Youngkwon Ko, Seok-Hwa Yoon, Yoon-Hee Kim, Sung-Mi Ji, Boohwi Hong

**Affiliations:** 1Department of Anesthesiology and Pain Medicine, Chungnam National University Hospital, Daejeon 35015, Korea; 2Department of Anesthesiology and Pain Medicine, College of Medicine, Chungnam National University, Daejeon 34134, Korea; 3Department of Anesthesiology and Pain Medicine, Dankook University Hospital, Cheonan 31116, Korea; 4Big Data Center, Biomedical Research Institute, Chungnam National University Hospital, Daejeon 35015, Korea

**Keywords:** cardiac output, hemodynamic monitoring, thermodilution, coronary artery bypass graft, arterial pressure waveform

## Abstract

(1) Background: Previous studies reported limited performance of arterial pressure waveform-based cardiac output (CO) estimation (FloTrac/Vigileo system; CO-FloTrac) compared with the intermittent thermodilution technique (CO_int_). However, errors due to bolus maneuver and intermittent measurements of CO_int_ could limit its use as a reference. The continuous thermodilution technique (CO_cont_) may relieve such limitations. (2) Methods: The performance of CO-FloTrac was retrospectively assessed using continuous recordings of intraoperative physiological data acquired from patients who underwent off-pump coronary artery bypass graft (OPCAB) surgery with CO monitoring using both CO-FloTrac and CO_cont_. Optimal time adjustments between the two measurements were determined based on R-squared values. (3) Results: A total of 134.2 h of data from 30 patients was included in the final analysis. The mean bias was –0.94 (95% CI, −1.35 to −0.52) L/min and the limits of agreements were −3.64 (95% CI, −4.44 to −3.08) L/min and 1.77 (95% CI, 1.21 to 2.57) L/min. The percentage error was 66.1% (95% CI, 52.4 to 85.8%). Depending on the time scale and the size of the exclusion zone, concordance rates ranged from 61.0% to 75.0%. (4) Conclusion: Despite the time adjustments, CO-FloTrac showed non-negligible overestimation, clinically unacceptable precision, and poor trending ability during OPCAB surgery.

## 1. Introduction

Arterial pressure waveform-based cardiac output (CO) estimation (Flotrac/Vigileo system; CO-FloTrac) has been widely used during the perioperative period. Despite the continued evolution of the algorithm (currently, fourth generation), previous studies reported limited performance during cardiac surgery [1,2,3,4]. 

As intermittent bolus thermodilution technique via pulmonary artery catheter (CO_int_) is considered the gold standard, devices for CO estimation are commonly compared with this method. However, CO_int_ is vulnerable to multiple sources of error such as volume, rate, timing, and temperature of the injectate [5]. Moreover, continuous tracking of CO via this method can be quite cumbersome.

Continuous thermodilution via pulmonary artery catheter (CO_cont_) can relieve such errors [6]. It continuously measures CO with a random on/off heating system [7]. However, as this technique is known to have innate time delay of about 4 to 12 min between measurement and actual CO [8,9,10], a direct comparison with other methods without considering this time delay may be misleading. 

Previous studies that evaluated CO-FloTrac have several limitations [1,2,3,11,12]. First, when CO_cont_ was used as a reference, the time delay between CO_cont_ and CO-FloTrac was not considered or not described explicitly. Second, when CO_int_ was used as a reference, the comparisons were made only intermittently (e.g., at prespecified time points or before and after an experimental maneuver), which may not reflect the entire clinical scene. This study, therefore, retrospectively compared CO-FloTrac and CO_cont_ (reference method) after time adjustments using continuous recordings of intraoperative physiological data during off-pump coronary artery bypass graft (OPCAB) surgery. 

## 2. Materials and Methods

### 2.1. Study Design

This retrospective study included consecutive patients who underwent OPCAB surgery with CO monitoring using both CO-FloTrac and CO_cont_ from March to December 2021. This study was performed as a sub-study of a project for developing noninvasive cardiac output estimation techniques (cris.nih.go.kr, accessed on 14 October 2022, KCT0006972). The study protocol was approved by the Institutional Review Board of Chungnam National University Hospital (CNUH 2021-04-090-006). Patients were excluded if their vital records did not include CO-FloTrac and CO_cont_; if a mal-positioned pulmonary artery catheter is noted by intraoperative central venous pressure waveform (e.g., right ventricular pressure waveform was noted instead) or postoperative chest radiography; if a persistent poor signal quality index (SQI = 4, which indicates severe problem with one or more aspects of the signal quality) on pulmonary artery catheter derived data is noted; or if known or persistent intraoperative arrhythmia is noted. Other data collected from their medical records included age, sex, weight, height, comorbidities, preoperative left ventricular ejection fraction, surgery type (conventional or minimally invasive), intraoperative fluid intake, transfusion, and use of vasopressor and/or inotrope infusion.

### 2.2. Data Acquisition and Preprocessing

All vital data were obtained from the prospective registry of vital signs for surgical patients at Chungnam National University Hospital (CNUH IRB 2019-08-039), which uses a free data collection program (Vital Recorder [13] version 1.8, accessed at https://vitaldb.net on 14 October 2022, Seoul, Korea). 

CO_cont_ data were acquired through Swan-Ganz catheter (7.5 F Swan-Ganz Continuous Cardiac Output Thermodilution Catheter: CCOmbo V, model 774F75, Edwards Lifesciences LLC) and a HemoSphere advanced monitoring platform (Edwards Lifesciences, Irvine, CA, USA) with CO measured and updated every 60 s (STAT mode, which displays a measurement without moving average process). 

CO-FloTrac data were acquired through an arterial catheter placed at the radial artery (brachial artery was used secondarily) and the FloTrac^TM^/Vigileo system (Edwards Lifesciences, Irvine, CA, USA) with CO estimated and updated every 20 s.

The collected data were extracted at a frequency of 1 Hz and filtered for extreme and/or clinically unplausible values so that systolic arterial pressure was ≤200 mmHg and ≥60 mmHg, diastolic arterial pressure was ≤110 mmHg and ≥30 mmHg, mean arterial pressure was ≤130 mmHg and ≥40 mmHg, pulse pressure (systolic–diastolic arterial pressure) was ≥20 mmHg, and SQI of the pulmonary artery derived signal was <4. CO-FloTrac values matched with extreme and/or clinically unplausible arterial pressures were also filtered. After these filtrations, values were averaged every 60 s so that both measurements (CO_cont_ and CO-FloTrac) could be matched at each minute. 

### 2.3. Time Adjustments

As there is a known time delay of CO_cont_ [8,10], and the degree of the delay can vary by case, time adjustment between CO_cont_ and CO-FloTrac was performed for each individual case. One to fifteen minutes of time-shifted values of CO_cont_ were matched with CO-FloTrac, and the optimal time adjustment was determined as the matching with the highest R-squared (R^2^) value between the two measurements. Each case was adjusted based on the determined optimal adjustment time. In cases with persistently low R^2^ values (<0.1) with any of the adjustments, 6 min of adjustment (based on previous studies [8,10]) was applied since no reasonable adjustment could be made based on such low correlation. 

### 2.4. Sensitivity Analysis

As the adjustment of time can affect the result, sensitivity analyses were performed to examine the extent to which results are affected by the adjustment. The analyses of main outcomes (bias, precision, and trending ability) were repeated after a fixed time adjustment and without any adjustment. The value for the fixed adjustment was determined based on previous findings. According to the previous studies, the in vitro response time to detect a 50% change in the CO_cont_ at 37 °C was 6.6 ± 0.8 min [8]. The in vivo response time to detect a 50% change in CO_cont_ at 37 °C lagged ultrasonic flowmeter by about 6 to 7 min [10]. Therefore, 6 minutes was used as the fixed value for the time adjustment in the sensitivity analysis. 

### 2.5. Statistics

The sample size was based on the available data during the study period. The correlation between CO_cont_ and CO-FloTrac was assessed by (1) determining R^2^ values and by (2) repeated measures correlation using the R package ‘rmcorr’, which considers non-independence of repeated measurements within individual cases [14]. Overall accuracy of CO-FloTrac was assessed by calculating root mean squared error (RMSE). The mean biases and the limit of agreements between time-adjusted CO_cont_ and CO-FloTrac were calculated using the R package ‘SimplyAgree’, which considers adjustment for repeated measurements per patient [15]. Percentage error was determined as a limit of agreement divided by the mean CO_cont_ of the cohort [16]. For a new CO estimation method to be accepted, it should have an equivalent precision of the reference. A combined precision of a reference and test method can be calculated as $\surd [{\left(precision\;of\;reference\right)}^{2}+\left(precision\;of\;test\;method{)}^{2}\right]$ [16]. As the pooled percentage error (i.e., combined precision) between continuous and intermittent thermodilution techniques was 29.7% in a recent meta-analysis [17] and the precision of the intermittent bolus thermodilution technique is commonly treated as 20% [16], we assumed the precision of the CO_cont_ to be 20% ($\sqrt{20^{2}+20^{2}}=28.3$). In this context, a percentage error of 30% was set as a cut-off for interchangeability in this study.

The trending ability of CO-FloTrac was assessed using four-quadrant plot analysis [18] with 3, 5, 10, and 20 min intervals for the change in the two methods. Two exclude zone sizes, less than 10% and 15% of CO change, were used for the calculation of the concordance rate (proportion of the points of first and third quadrants of the four-quadrant plot). All statistical analyses were performed using R software version 4.0.3 (R Project for Statistical Computing, Vienna, Austria).

## 3. Results

A total of 36 cases were assessed for eligibility and six were excluded due to persistent high SQI (*n* = 3), inappropriate catheter tip position (*n* = 2), and persistent intraoperative arrhythmia (*n* = 1) (Figure 1). Finally, 134.2 h of data from 30 patients was included in the final analysis. The clinical characteristics of the included patients are summarized in Table 1.

The R^2^ between varying degrees of time-adjusted CO_cont_ and CO-FloTrac are shown in Figure 2. Repeated measures correlation and RMSE between the two measurements before and after the time adjustment (determined by the highest R^2^ value) are shown in Figure 3. A moderate degree of correlation was noted, and it slightly increased from 0.479 to 0.582 after the adjustment. Meanwhile, RMSE was minimally changed by the adjustment. A sample case showing the matching between CO_cont_ and CO-FloTrac before and after the adjustment is shown in Figure 4.

The results of Bland–Altman analysis are shown in Figure 5. The mean bias (CO_cont_—CO-FloTrac) was −0.94 (95% CI, −1.35 to −0.52) L/min and the limits of agreement were −3.64 (95% CI, −4.44 to −3.08) L/min and 1.77 (95% CI, 1.21 to 2.57) L/min. The percentage error was 66.1% (95% CI, 52.4 to 85.8%).

Four-quadrant plots using 3 to 20 min interval changes of CO_cont_ and CO-FloTrac are shown in Figure 6. Depending on the time scale and the size of the exclusion zone, concordance rates ranged from 61.0% to 75.0%.

The results of the sensitivity analysis are summarized in Appendix A. The performance of CO-FloTrac assessed in the dataset with a six-minute time adjustment showed minimal difference from the results of the dataset with individual time adjustments. On the contrary, a slight increase in percentage error (68.2%) and considerable compromise in the trending ability (concordance rate, 48.2% to 66.1%) were noted in the dataset without time adjustment.

## 4. Discussion

This study assessed the accuracy, precision, and trending ability of CO-FloTrac in the OPCAB cohort after adjusting CO_cont_ values for the time delay. Despite the individually optimized time adjustments, the precision of CO-FloTrac was clinically unacceptable (percentage error of 66.1%), and the trending ability was poor. To the best of our knowledge, no study has compared CO estimations continuously during the entire operative period after adjusting the time delay in the reference method.

Currently, the gold standard method for CO measurement is CO_int_. However, as the technique includes a manual bolus maneuver, it is not free from errors due to manipulations and is not suitable for continuous recording during dynamic intraoperative periods. The use of CO_cont_ could relieve such limitations [6,7].

The percentage error of CO-FloTrac shown in this study was consistent with previous findings [1,3,4]. Jeong et al. reported −0.23 L/min of bias (overestimation of CO-FloTrac) and 57% error between CO-FloTrac (version 1.10) and CO_cont_ (unadjusted) [12]. Most other studies used CO_int_ as a reference and reported −0.66 to 0.05 L/min of bias [1,2,19] and 33.8% to 56.8% of error [1,2,3,4,19]. The inconsistent and unreliable performance of the FloTrac/Vigilleo system might be partly explained by vascular tone or pulse pressure, as previously reported [1,2,20,21]. It seems that in a small portion of this cohort, however, CO-FloTrac showed a fair precision and trending ability during the entire case (e.g., the sample case in Figure 4) despite dynamic changes in hemodynamics and inevitable accompanying changes in vascular tone. Factors contributing to this conservation of CO-FloTrac performance should be sought in future studies. Further understanding of conditions for better performance of CO-FloTrac may enhance its clinical applicability and complement the current limitations of CO_cont_ such as time delay and invasiveness.

Although the STAT mode of CO_cont_, the reference method in the current study, was introduced to provide more prompt clinical guidance, it is still not free from the response time and it was clearly shown in previous animal [10], in vitro [8], and clinical [9] studies. To overcome this technical challenge, we adopted a time shifting strategy. As seen in the sample case (Figure 4), we tried to make the best time adjustment between real-time estimation (CO-FloTrac) and delayed measurement (CO_cont_) and match their peaks and troughs optimally. As shown in the sensitivity analysis, the time adjustments, using either a fixed value or individually determined values, improved the performance of CO-FloTrac, especially the trending ability. However, as this process was based on the metrics (R^2^) calculated from an entire recording from each individual dataset, it could not match the peaks and troughs completely throughout the entire period. In other words, although the adjustment could correct the inter-individual variability of time delay, it could not explicitly correct the intra-individual variability.

A distinctive characteristic of the current study compared with the previous ones is that the time dimension is considered in detail for the assessment of trending ability. It is obvious that metrics such as concordance rate can differ by the selection of time scale. However, the time scales used in previous studies were too broad (e.g., interval between the administration of protamine and the start of sternal closure) [2,22] or not explicitly described (e.g., simply before and after a certain maneuver) [3,11]. To properly interpret and apply the results to actual clinical practice, the time scale should be considered in detail. Additionally, the size of the exclusion zone can affect the concordance rate, as shown in this study. Thus, care is needed in comparing the results between studies using different exclusion zone cut-offs.

This study has several limitations. First, as the study includes physiological signals continuously acquired during the entire operative period, some low quality data could have been involved in the analysis despite the filtrations. Second, there are inherent limitations due to the retrospective nature, including a lack of detailed information regarding vasopressor and/or inotrope uses (e.g., dose adjustments and administration time points) and rapid fluid infusions which can affect the fidelity of the measurements. Third, no statistical power analysis was performed. Fourth, the approach using time adjustment for CO_cont_ was unprecedented, or at least not widely used in this field. However, we believe that this adjustment should be considered in the assessment of CO, and the amount of adjustment is plausible according to the previous studies [8,9,10].

## 5. Conclusions

In conclusion, despite the time adjustments, the estimation of CO using the FloTrac/Vigileo system showed non-negligible overestimation (mean bias –0.94 L/min), clinically unacceptable precision (percentage error 66.1%), and poor trending ability compared with the continuous thermodilution technique as a reference during OPCAB surgery.

## Figures and Tables

**Figure 1 jcm-11-06093-f001:**
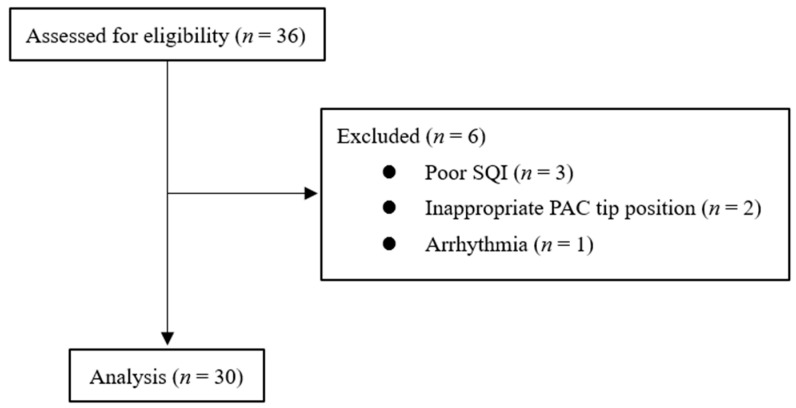
Patient flow diagram. SQI: signal quality index, PAC: pulmonary artery catheter.

**Figure 2 jcm-11-06093-f002:**
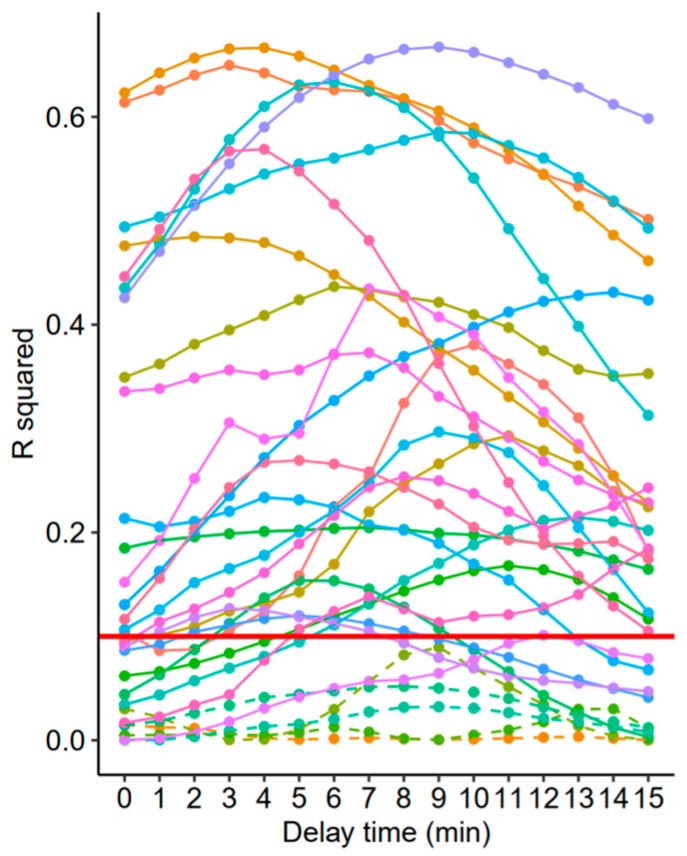
R-squared (R^2^) values between arterial pressure waveform-based cardiac output estimation (CO-FloTrac) and varying degrees of leading values (values of antegrade time points; 1 to 15 min) of continuous thermodilution cardiac output (CO_cont_). The optimal time adjustment was defined as the delay when R^2^ was highest within an individual dataset. In cases with persistently low R^2^ values (<0.1) with any of the adjustments, 6 min of adjustment was applied since no reasonable adjustment could be made based on such low correlation. Each color represents each individual data. The red solid horizontal line indicates R^2^ = 0.1. Five cases with persistently low R^2^ values are indicated as dashed lines.

**Figure 3 jcm-11-06093-f003:**
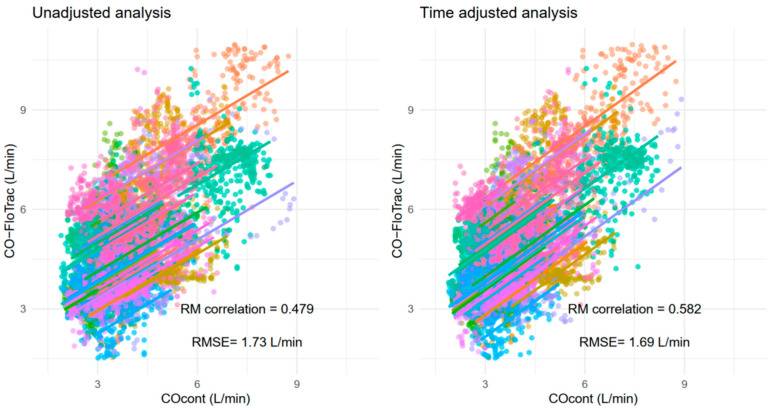
Repeated measures correlation (RM correlation) and root mean squared error (RMSE) between arterial pressure waveform-based cardiac output estimation (CO-FloTrac) and continuous thermodilution cardiac output (CO_cont_) before and after time adjustment. Each color represents each individual dataset.

**Figure 4 jcm-11-06093-f004:**
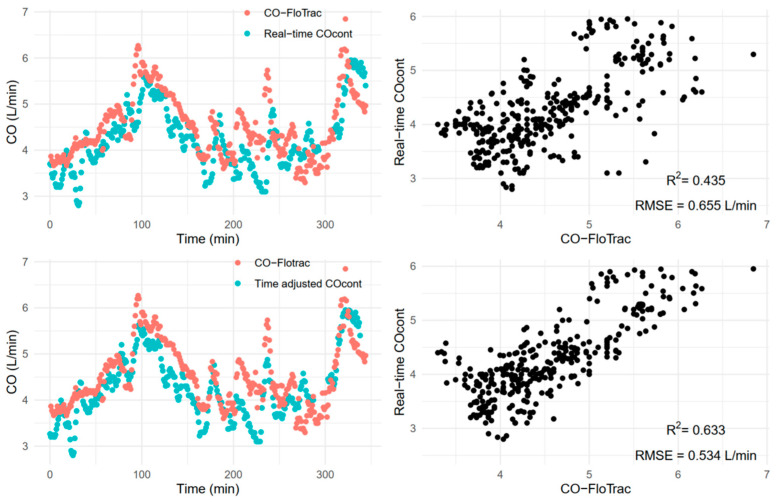
Continuous thermodilution cardiac output (CO_cont_) and arterial pressure waveform-based cardiac output estimation (CO-FloTrac) of a sample case before (**upper panel**) and after the time adjustment (**lower panel**). The plots on the left side are time-series plots and the plots on the right side are scatter plots between the two variables. Note the slightly misaligned peaks and troughs due to the time delays in the CO_cont_ values (**left upper**). The plot on lower left side shows a more optimized alignment after the time adjustment (in this case, 6 min). R-squared (R^2^) value increased from 0.435 to 0.6333 and root mean squared error (RMSE) decreased from 0.655 to 0.534 L/min after the time adjustment.

**Figure 5 jcm-11-06093-f005:**
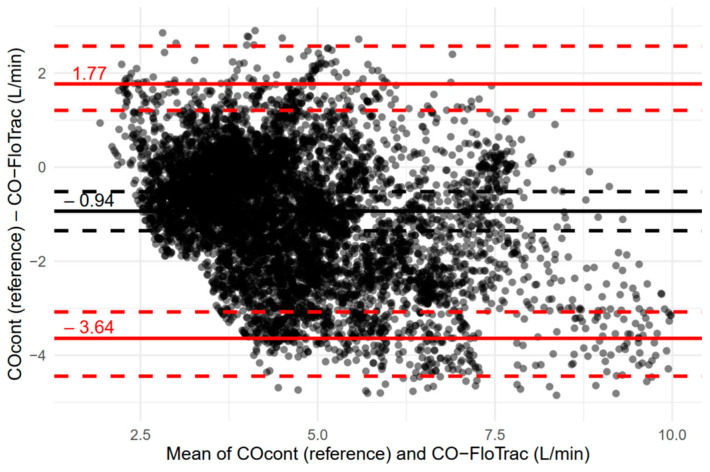
Bland–Altman plot for time-adjusted continuous thermodilution cardiac output (COcont, reference method) and arterial pressure waveform-based cardiac output estimation (CO-FloTrac; FloTrac/Vigileo system). Black and red solid lines indicate mean bias and limits of agreement, respectively. Dashed lines indicate 95% confidence intervals of the corresponding parameters. The negative mean bias presented in the figure indicates the overestimation of CO-FloTrac, and 95% of the biases (difference between the two measurements) reside within the presented limits of agreement.

**Figure 6 jcm-11-06093-f006:**
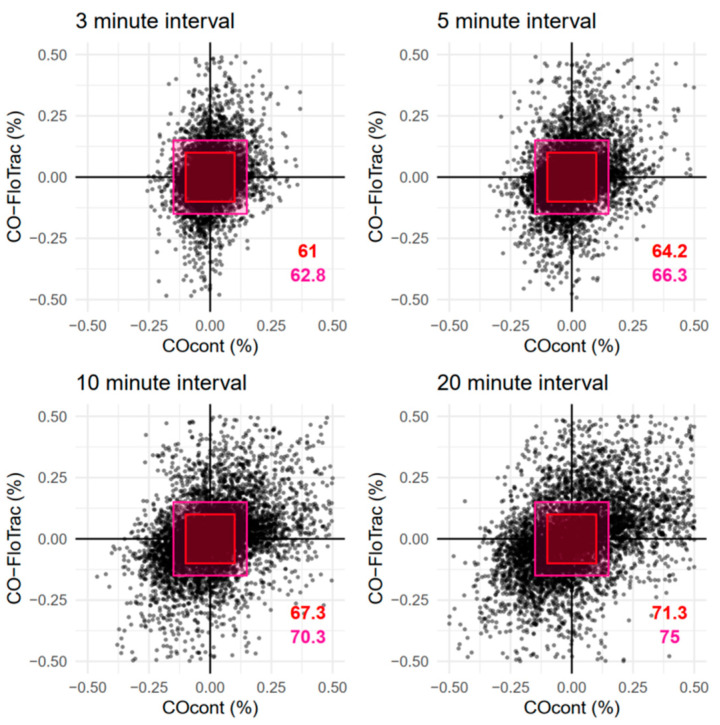
Four-quadrant plot for 3 to 20 min interval change of continuous thermodilution cardiac output (CO_cont_) and arterial pressure waveform-based cardiac output estimation (CO-FloTrac). Each axis indicates interval change of the corresponding parameter. Red rectangles (inner box) indicate the exclusion zone of 10% and deep pink rectangles (outer box) indicate the exclusion zone of 15%. Colored numbers indicate corresponding concordance rates.

**Table 1 jcm-11-06093-t001:** Clinical characteristics.

Characteristics	Value
Age (years)	65.0 ± 10.6
Sex (F/M)	5/25
Height (cm)	164.2 ± 8.2
Weight (kg)	68.7 ± 11.2
BMI (kg/m^2^)	25.4 ± 3.0
Comorbidities	
• DM	18 (60.0)
• HTN	20 (66.7)
• Chronic kidney disease	4 (13.3)
■ No hemodialysis	2 (6.7)
■ On hemodialysis	2 (6.7)
Left ventricular ejection fraction (%)	
• ≥40	26 (86.7)
• <40	4 (13.3)
Surgery type	
• Conventional	22 (73.3)
• Minimally invasive	8 (26.7)
Intraoperative fluid intake	
• Crystalloid (mL)	2943.8 ± 1342.1
• Colloid (mL)	0.0 (0.0, 500.0)
Transfusion (mL) *	773.0 (248.0, 1386.0)
• PRBC (unit)	1 (0, 3)
• FFP (unit)	0 (0, 0)
• Salvaged blood (mL)	304.5 (100.0, 1032.0)
Vasopressor infusion	28 (93.3)
Inotrope infusion	21 (70.0)
Recorded time (h)	5.7 ± 1.4
CO_cont_ (L/min)	3.8 (3.2, 4.7)
CO-FloTrac (L/min)	4.8 (3.8, 6.1)

Values are count (%), mean ± SD, or median (IQR). * Includes packed red blood cells (PRBC), fresh frozen plasma (FFP), and salvaged blood volumes. BMI: body mass index, DM: diabetes mellitus, HTN: hypertension, CO_cont_: continuous thermodilution cardiac output, CO-FloTrac: arterial pressure waveform-based cardiac output estimation (FloTrac/Vigileo system).

## Data Availability

The data presented in this study are available on reasonable request from the corresponding author.

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
