# Peer review of "Comparison between Fourth-Generation FloTrac/Vigileo System and Continuous Thermodilution Technique for Cardiac Output Estimation after Time Adjustment during Off-Pump Coronary Artery Bypass Graft Surgery: A Retrospective Cohort Study"

_jcm, 2022, doi:10.3390/jcm11206093_

Round 1

Reviewer 1 Report

Materials and methods:

Please give defininition on signal quality indicicators/index.

„Other data collected from their medical records included patient age, sex, weight and height.” means no additonal information since these should be inserted before any CO calculations.

„Patients were excluded if…known or persistent intraoperative arrhythmia is noted.” Please explain how to implement these methods for those patients ondergoing OPCAB with arrhythmias e.g. atrial fibrillation that is the most common non-malignant arrhythmia in the elderly.

”…CO estimated and updated every 60 seconds… without moving average process…” How may Authors compare this sort of sampling method to the other CO-FloTrac.

Lines 88-96 should be clarified, explained and rephrased due to misleading definitions of unplausible data of <200 mmHgof sytolic blood pressure and so on.

Results:

Fig 2.:  Most of the curves are above the low R2 <0.1, see p.3/10, line 104. Please explain the figure in more details, since in this format it gives no acceptable conclusion.

Fig 3.: „repeated measures of correlation between the 2 measurements was 0.569”. Please indicate how it may be implemented in the real life since this result is a bit higher than flipping up a coin.

Fig. 4.: Please indicate the use and dosage of inotropes and vasopressors during the measurements. These may influence of the data of FloTrac measured at the extremities due to increasing cardiac performance and centralizing the circulation. By sight there is not much difference between the Real time and the Time adjusted COcont. please give numeric values to define the difference.

Fig 5. Legends refer to dotted lines but there is no any dottted line in the figure. Bland Altman plot should have a reference. 

„…slight decrease in precision (68.2% error)”. Please state waht do you mean on the near 70% of error as slight decrease. The Conclusion mentions another value by the way…

Author Response

Reviewer 1.

Please give definition on signal quality indicicators/index.

The definition of SQI (as described in the manufacturers instruction) was added in the first appearance of the term in the manuscript as follows:

“persistent poor signal quality index (SQI=4, which indicates severe problem with one or more aspects of the signal quality)”

“Other data collected from their medical records included patient age, sex, weight and height.” means no additonal information since these should be inserted before any CO calculations.

As you mentioned, this information is routinely inserted into the monitoring device to determine calibration factors or derive an indexed value of cardiac output (i.e., cardiac index). Unfortunately, however, this information cannot be extracted from our recording system (VitalRecorder) and should be extracted from the electronic medical record. Besides, we did not use the indexed value for the comparison in the study since the original data derived from the continuous thermodilution technique is not an indexed value. 

During the revision process, we extended the extraction of clinical information as follows:

“Other data collected from their medical records included age, sex, weight, height, comorbidities, preoperative left ventricular ejection fraction, surgery type (conventional or minimally invasive), intraoperative fluid intake, and use of vasopressor and/or inotrope infusion.”

“Patients were excluded if…known or persistent intraoperative arrhythmia is noted.” Please explain how to implement these methods for those patients ondergoing OPCAB with arrhythmias e.g. atrial fibrillation that is the most common non-malignant arrhythmia in the elderly.

The purpose of the current study was not to insist on or validate the use of CO-FloTrac in patients with arrhythmia. Proper assessment of the performance of CO-FloTrac during OPCAB surgery was the primary goal of the study.

As the arterial pressure waveform device (FloTrac) estimates cardiac output based on the characteristics of the waveform, the validity or reliability of this technique in patients with arrhythmia has not been firmly established, or at least not recommended to be used in such patients. Although arrhythmia is not an absolute limiting factor for the FloTrac system according to the manufacturer, we empirically do not rely on the system during periods with irregular rhythm since the value frequently changes. Also, most of the previous studies of FloTrac commonly excluded cases with arrhythmia. 

On the other hand, continuous thermodilution technique is free from such limitations. As our study aimed to compare these methods, the prerequisites for the reliability of both methods should be met. 

“…CO estimated and updated every 60 seconds… without moving average process…” How may Authors compare this sort of sampling method to the other CO-FloTrac.

There are two kinds of CO estimates from the continuous thermodilution technique. One is Trend mode CO which is a moving averaged value of STAT mode CO. Therefore, Trend mode CO would be affected by the former 5 to 10 minute CO values. Thus, STAT mode CO is a more reasonable value to be compared with CO-FloTrac. Also, as STAT mode CO represents CO of the 1 minute interval, only selectively using the latest 20 seconds CO-Flotrac value seems unreasonable. Therefore, we compared STAT mode CO and 1-minute averaged CO-FloTrac value in the current study. 

Lines 88-96 should be clarified, explained and rephrased due to misleading definitions of unplausible data of <200 mmHgof sytolic blood pressure and so on.

We respectfully guess there is some misunderstanding. Indeed, the described cut-offs are the same as exclusion criteria, the description is intended for inclusion criteria (please note “so that” in the description).

“The collected data were extracted at a frequency of 1 Hz and filtered for extreme and/or clinically unplausible values so that systolic arterial pressure was ≤200 mmHg”

Results:

Fig 2.: Most of the curves are above the low R2 <0.1, see p.3/10, line 104. Please explain the figure in more details, since in this format it gives no acceptable conclusion.

Thank you for the comment. We found that there are some cases that show a poor correlation between COcont and CO-FloTrac despite the various attempts for the time adjustment. In such cases, it may be misleading to make a time adjustment between the two measurements based on the low correlation. Therefore, no adjustment was done in such cases. We added a brief description in the figure note as follows:

“In cases with persistently low R2 values (<0.1) with any of the adjustments, no adjustment was applied since no meaningful adjustment could be made based on such low correlation.”

Fig 3: “repeated measures of correlation between the 2 measurements was 0.569”. Please indicate how it may be implemented in the real life since this result is a bit higher than flipping up a coin.

Repeated measures of correlation is a kind of metric for correlation that quantifies the association between the two continuous variables (range from -1 to 1). Although it may seem arbitrary, we can say that there is a moderate degree of correlation (or association) between the two measurements if the correlation coefficient is 0.25 to 0.75. It is, therefore, hardly conceived as a binary concept (all or nothing, flipping coin). We can expect one variable would increase when the other variable increases (with a moderate degree of association). We clarified this meaning by revising the description in the result as follows:

“A moderate degree of correlation was noted, and it was slightly increased from 0.479 to 0.569 after the adjustment.”

As you mentioned, the relation between the two measurements was not perfectly linear (correlation coefficient near 1). This fact per se may indicate CO-FloTrac is not a good measure for CO (COcont as the reference). However, it is only a gross measure of comparison between the two methods, and it cannot replace the standard (recommended) method of comparison for continuous measures, which is the Bland-Altman analysis.

Based on the results, we concluded that CO-FloTrac is not a good measure for CO during OPCAB surgery. And we believe that this intention of the authors was adequately described in the manuscript (a large amount of bias, poor precision, poor trending ability in the main and sensitivity analyses).

Fig. 4.: Please indicate the use and dosage of inotropes and vasopressors during the measurements. These may influence of the data of FloTrac measured at the extremities due to increasing cardiac performance and centralizing the circulation. 

Thank you for the comment. As you mentioned, those factors could surely affect the contour of the arterial pressure waveform and so on CO-FloTrac value. And we reason that those are one of the factors that hampered the performance of CO-FloTrac in OPCAB surgery where such medications are frequently used. Unfortunately, however, we could not extract the exact dose, administration point, and dose adjustments for those drugs. Alternatively, we extracted a schematic version of such information (use or not) and added it to Table 1.

By sight there is not much difference between the Real time and the Time adjusted COcont. please give numeric values to define the difference.

Thank you for the suggestion. To show the quantity of improvement in the comparison by the time adjustment, we revised Figure 3 and Figure 4. Additionally, root mean squared error (RMSE) was used for the quantification.

“Repeated measures correlation and RMSE between the two measurements before and after the time adjustment (determined by the highest R2 value) are shown in Figure 3. A moderate degree of correlation was noted, and it slightly increased from 0.479 to 0.569 after the adjustment. Meanwhile, RMSE was minimally changed by the adjustment. A sample case showing the matching between COcont and CO-FloTrac before and after the adjustment is shown in Figure 4.” 

Fig 5. Legends refer to dotted lines but there is no any dottted line in the figure. Bland Altman plot should have a reference. 

Thank you for the correction. Terminology was corrected accordingly. We are not sure what “reference” meant in your comment. It may indicate the reference method or reference line (zero bias line). We revised the figure and indicated both (reference method or zero bias line) in it.

„…slight decrease in precision (68.2% error)”. Please state waht do you mean on the near 70% of error as slight decrease. The Conclusion mentions another value by the way…

Sorry for the confusion. We respectfully guess that the confusion is due to the multiple similar outcomes in the main result and sensitivity analysis. 

In the main analysis (individually time adjusted dataset), the percentage error of CO-FloTrac was 66.3%. In the non-adjusted dataset (sensitivity analysis), it was 68.2%. As these values are a measure of error, a slight increase from 66.3% to 68.2% indicates a slight decrease in precision (increase in error = decrease in precision). Meanwhile, the trending ability was considerably compromised in the non-adjusted dataset from 58.4-74.0% to 48.2 to 66.1%. 

To minimize misunderstanding, we revised the description as follows:

“On the contrary, a slight increase in percentage error (68.2%) and considerable compromise in the trending ability (concordance rate, 48.2 to 66.1%) were noted in the dataset without time adjustment.”

Based on our main analysis, 66.3% of percentage error was presented in the conclusion. 

Reviewer 2 Report

The authors report that Co-FloTrac showed overestimation compared to COcont despite time adjustment. The poor correlation between CO estimation using FloTrac devices and CO measured using the Swan-Ganz catheter have been previously demonstrated in literature, but the time adjustment method brings a novel insight. The study has inherent limitations related to its retrospective nature and the lack of statistical power analysis and this should be detailed in limitations. CO is not estimated, but measured using the Swan-Ganz catheter. Over or underestimations of the measured values are related to various sources of error, technical or related to cardiac pathologies. Please consider rephrasing. In the methods section, it is said that "malpositioned PAC is noted by intraoperative central venous pressure waveform" - please consider rephrasing, as PAC tip malposition is not identified by central venous pressure waveform. Please consider extending the existing literature review in the discussion section regarding comparisons between FloTrac and Swan-Ganz performance.

Author Response

Reviewer 2.

The authors report that Co-FloTrac showed overestimation compared to COcont despite time adjustment. The poor correlation between CO estimation using FloTrac devices and CO measured using the Swan-Ganz catheter have been previously demonstrated in literature, but the time adjustment method brings a novel insight. The study has inherent limitations related to its retrospective nature and the lack of statistical power analysis and this should be detailed in limitations. 

Thank you for the comment. Those limitations were added as follows:

“Second, there are inherent limitations due to the retrospective nature, which includes lack of detailed information regarding vasopressor and/ or inotrope uses (e.g., dose adjustments and administration points) and rapid fluid infusions which can affect the fidelity of the measurements. Third, no statistical power analysis was done.”

CO is not estimated, but measured using the Swan-Ganz catheter. Over or underestimations of the measured values are related to various sources of error, technical or related to cardiac pathologies. Please consider rephrasing. 

Thank you for the correction. The terminology was corrected throughout the entire manuscript accordingly.

In the methods section, it is said that "malpositioned PAC is noted by intraoperative central venous pressure waveform" - please consider rephrasing, as PAC tip malposition is not identified by central venous pressure waveform. 

According to the manufacturer’s manual (from Edward), the thermistor (4 cm proximal to the catheter tip) should be in the main body of the pulmonary artery. Also, the thermal filament for COcont should rest between RA and RV. Erroneous CO measurement may result if this filament resides beyond the pulmonic valve. Based on these recommendations, abnormal CVP waveform (e.g., RV pressure waveform) can signal a mal-positioned PAC tip. Indeed, there was one case that showed persistent RV pressure waveform from CVP input. A postoperative chest radiograph revealed that the catheter had been advanced too deeply and the catheter is kinked at the distal pulmonary artery. As the position of both the thermistor and thermal filament was determined to be too deep, the case was excluded from the study. To suggest this rationale, the exclusion criterion was elaborated as follows:

“mal-positioned pulmonary artery catheter is noted by intraoperative central venous pressure waveform (e.g., right ventricular pressure waveform was noted instead) or postoperative chest radiography”

Please consider extending the existing literature review in the discussion section regarding comparisons between FloTrac and Swan-Ganz performance.

Thank you for the suggestion. We further extended the review regarding this issue in the discussion.

Reviewer 3 Report

This is an exceptionally good retrospective study which aims at comparing between fourth generation FloTrac/ Vigileo system and continuous thermodilution technique for cardiac output estimation after time adjustment during off-pump coronary artery bypass graft surgery.

It is an interesting topic that will be of interest to the readers of the journal. Generally, the article is well constructed, very well written and structured.

I have one comment regarding figure legends. Figure titles are self-descriptive but few lines within the manuscript should be included withing the figure legends, i.e.

Lines 183-184 to be the figure legend for Figure 3

Lines 192-193: “The plots on the upper panel show CO FloTrac matched with unadjusted (real-time) COcont values” and lines 194-195: “The plots on the lower panel 194 show more optimized alignment after time adjustments. (6 and 5 minute for case #16 and 195 #12, respectively” should be included in Figure 4 legend.

Lines 206-207: included in Figure 5 legend

Lines 216-218: included in Figure 6 legend

The smaller font size of lines 206-207 and 216-218 indicates that these are probably figure legends but a space added inadvertently.    

Author Response

Thank you for the invaluable corrections. The manuscript was revised accordingly.

Reviewer 4 Report

This is an interesting, retrospective analysis by Oh et al. about thermodilution techniques after CABG.

I have no further suggestions other than the fact that more baseline characteristics about included patients would be good. Also, as mentioned in the limitations section, the use of catecholamines/inotropes/albumin would be interesting.

Author Response

Reviewer 4.

This is an interesting, retrospective analysis by Oh et al. about thermodilution techniques after CABG.

I have no further suggestions other than the fact that more baseline characteristics about included patients would be good. Also, as mentioned in the limitations section, the use of catecholamines/inotropes/albumin would be interesting.

Thank you for the comment. We further extracted clinical information (comorbidity, preoperative LV ejection fraction, intraoperative use of vasopressor and inotrope infusion, intraoperative fluid intake, surgery type) and revised the Table 1.

Round 2

Reviewer 1 Report

Page 2, Line 64: „This retrospective study included consecutive patients who underwent OPCAB… „Why the word „consecutive” was deleted? This means increase in the enrollment bias of the study.

Page 3, Lines 109-111.: „In cases with persistently low R2 values (<0.1) with any of the adjustments, no adjustment was applied since no meaningful adjustment  could be made based on such low correlation. In cases with persistently low R2 value (<0.1) with any of the adjustments, no adjustment was applied.” What was the action taken? Were there any data deleted?

Page 3, Lines 109-111.: „In cases with persistently low R2 values (<0.1) with any of the adjustments”, no meaning of this is visible. Please indicate the real clinical value .

Page 3 line 127. Does th e R2 mean R2 ?

Page 4 fig 1. should be changed. Eligible cases n=36 should be defined as screened cases ,total. „Analysis (n=30)” group is the „ eligible cases” group.

Page 5 table 1. cont. Please state separately the units and volumes of the units of the blood components transfused. 

Page 6. fig 2. At least 11 of the patients R2 <0.1. Please indicate and expain avoidance of  the statistical bias.

Page 8. Fig 4. is a bit confusing. Upper part shows a better result after adjustment in Case #16 and #12, but I hardly find the definitions and explanation of the 4 plots in the lower part of the Figure. Please clarify.

Page 9, Fig 5. „Black and red solid lines indicate mean bias and limits of agreement, respectively. Dotted Dashed lines indicate 95% confidence intervals of the corresponding parameters. Black dotted line indicates zero bias.”. Please detail more thoroughly, since the legend is hard to understand and analyze.

Author Response

Thank you for the thorough review. 

Please find our response to your review attached below.
